# Transcriptome survey and expression analysis reveals the adaptive mechanism of 'Yulu Xiang' Pear in response to long-term drought stress

**Sheng Yang, Mudan Bai, Guowei Hao, Xiaowei Zhang, Huangping Guo** [ORCID]**\*, Baochun Fu\***

Pomology Institute, Shanxi Agricultural University, Shanxi Key Laboratory of Germplasm Improvement and Utilization in Pomology, Taiyuan, Shanxi, PR China

\* ghping1959@163.com (HG); 397999001@qq.com (BF)

## Abstract

Pear is one of the most important economic fruits worldwide. The productivity is often negatively affected by drought disaster, but the effects and adaptive mechanism of pear in response to drought stress has not been well understood at the gene transcription levels. Using Illumina HiSeq 2500, the transcriptome from 'Yulu Xiang' Pear leaves were sequenced and analyzed to evaluate the effects of long-term drought stress on the expression of genes in different biosynthetic pathways. Results showed that long-term drought stress weakened antioxidant systematization and impaired the synthesis of photosynthetic pigment in 'Yulu Xiang' Pear leaves. The reduced light utilization and photosynthetic productivity finally resulted in the inhibited fruit development. The transcriptome survey and expression analysis identified 2,207 differentially expressed genes (DEGs) which were summarized into the 30 main functional categories. DEGs analysis showed that the enzyme genes involved in phenylpropanoid biosynthesis under drought stress were up-regulated, and the promoted process of phenylpropanoid synthesis may be beneficial to reduce the transpiration rate and increase water use efficiency of 'Yulu Xiang' Pear leaves. Up-regulated malate dehydrogenase expression were also observed in drought stress groups, and the activated soluble sugar biosynthesis could be helpful to promote osmotic regulation and increase antioxidant capacity to enhance drought resistance of leaves. The mRNA expression of enzyme genes associated with hormones including ethylene, abscisic acid, and gibberellin were higher in drought stress groups than that in control, indicating a promoted cell proliferation under drought stress. Long-term drought stress significantly decreased photosynthetic productivity, and negatively affected development of 'Yulu Xiang' Pear. Transcriptome survey and expression analysis reveals that the inhibited photosynthesis could be closely related with drought-induced lignification and hormones synthesis, and the present dataset can provide more valuable information to analyze the function of drought stress-related genes improving plant drought tolerance.

**Data Availability Statement:** All relevant data are within the paper.

**Funding:** This work was partially supported by the China Agriculture Research System (CARS-28-28),

The Research Subject of Agricultural Science and Technology Innovation of Shanxi Academy of Agricultural Sciences (YCX2018D2YS14), Shanxi Province Natural Science Foundation (201801D121255), The Project of Scientific and Technological Innovation research of Shanxi Academy of Agricultural Sciences (YCX2020SJ10).

**Competing interests:** The authors have declared that no competing interests exist.

## Introduction

Pear is one of the most widely grown fruit species worldwide, but the unpredictable reduction from adverse weather (such as drought disaster) is an urgent problem [1, 2]. The drought disaster is a uncertainty and dangerous climatic hazard threatening fruits production in many areas of the world [3–5]. Plants is inseparable from water, and lack of water can significantly inhabit the normal physiological function, such as photosynthesis. Studies found that water deficiency was second only to the shortage of illumination causing the reduced photosynthetic rate [6, 7]. Water deficiency may promote transpiration by increasing temperature of plant leaves, and the contradiction between supply of water and demand of plant body may result in the reduced photosynthetic rate. Maes et al. [8] reported that the water potential, relative water content, leaf area, stomatal conductance of fruit seedlings leaves under drought stress was reduced, which finally caused the decline of leaf photosynthetic rate. Down-regulated stomatal conductance induced by drought stress inhibited the entry of carbon dioxide from the external environment into leaves, and increased resistance of mesophyll cells, and weakened carboxylase activity [9].

The above effects may result from the plant adaptation to drought stress, and it is necessary to clear mechanism of plant tolerance on drought stress. Studies found that drought stress related genes alterations were induced by plants perceptived and transmited drought signal. Many molecular components responsive to drought stress have been identified, and these stress-responsive genes were involved in intracellular osmotic pressure, transcriptional regulation, protein phosphatases, and antioxidant system. These suggest that responses of plants on drought stress is a complex process and connection, but the highly complex and interconnected network is still not understood. Sequencing of the pear genome provides a possibility for gene prediction and annotation. 'Yulu Xiang' Pear, one the most widely fruit in China, were chosen in our present study. Drought stress treatment groups and normal watering control groups were set up, respectively. Transcriptome sequencing and analysis techniques were used to reveal the regulation mechanism of drought stress on the growth and development of 'Yulu Xiang' Pear at the transcriptome level. The knowledge about the expression patterns of genes in 'Yulu Xiang' Pear under drought stress may be helpful in narrowing down on candidate genes, which can be used as a foundation to improve the quality of 'Yulu Xiang' Pear during cultivation in breeding in the future.

## Materials and methods

### Plant material and growth conditions

'Yulu Xiang' Pear were investigated in pear germplasm resources of Institute of Fruit (Shanxi Academy of Agricultural Science) located in the southwest of Taigu County, Shanxi Province (latitude 37˚23'0"N, longitude 112˚32'0"E). The garden is a warm temperate continental climate with an average annual temperature of 10.6˚C and sunshine hours of 2300 hours/year, and annual precipitation and evaporation of is about 450 and 1800 mm/year, respectively.

### Test design

The 10-year-old healthy 'Yulu Xiang' Pear trees were used in the present research, and the test was carried out during October 21, 2017 to October 21, 2018. Pear trees of control groups were watered for four times in October (2017), March (2018), May (2018) and July (2018), respectively. For drought treatment groups, there was no water spray in the orchard. The soil moisture content was detected by the method of weighting and replenishing water after 12 months of plant growth, and the soil relative water content of pear trees cultivation in control groups and drought stress groups were 80% and 30%, respectively.

## Detection of photosynthetic characteristics, chlorophyll content, soluble solids, antioxidant indicators, malondialdehyde (MDA) and osmoregulation substance in 'Yuluxiang' Pear leaves

After long-term drought treatment, photosynthetic indexes including net photosynthetic rate (Pn), stomatal conductance (Gs), intercellular $CO_2$ concentration (Ci) and transpiration rate (Tr) were measured by LI-6400 Portable Photosyntometer ((Licor, USA)). Chlorophyll was extracted using 95% ethanol, and the contents of chlorophyll a, chlorophyll b and carotenoids in the extract were determined according to the method of Li et al. [10]. The soluble solids was measured by the PAL-1 digital display sugar meter. The content of malondialdehyde (MDA) was determined by thiobarbituric acid method (TBA), the activity of superoxide dismutase (SOD) was determined by nitrogen blue tetrazole reduction method, the activity of peroxidase (POD) was determined by guaiacol method, and the activity of catalase (CAT) was determined by ultra-violet absorption method. The content of soluble protein was determined by Coomassie brilliant blue method, and the content of proline was determined by acid ninhydrin method. There were six replicates for each treatment and the above experiments were repeated three times.

## Total RNA extraction, cDNA library construction, and deep sequencing

Total RNA in six replicates leaf samples from different pear trees for each treatment were isolated following the protocols of Trizol (Invitrogen, Carlsbad, CA, USA). RNA content was detected by ND2000 (NanoDrop 2000, Thermo., USA), and RNA samples that had 260/280 ratios above 2.0 were used for subsequent experiments. The sequencing library was constructed by Personal Biotechnology Co., Ltd. (Shanghai, China) via an Illumina MiSeq instrument., and the transcriptome double terminal sequencing was performed by using Illumina hiseq 2500 sequencing platform (Illumina, CA, USA). Clean reads was obtained after that the low-quality reads were discarded by using the Seq-Prep program (https://github.com/jstjohn/SeqPrep), and the clean reads were further de novo assembled by Trinity software [11].

## Differentially expressed genes (DEGs) and functional annotation

The RPKM method (reads per KB per million reads) was used to calculate the expression level of genes. The differential expressed genes (DEGs) were defined as the genes with false discovery rate (FDR) $P<0.05$ and multiple difference $\geq 2$. Gene function was annotated by using BlastX with an E value less than $10^{-5}$ against NCBI non-redundant (Nr) databases (http://www.ncbi.nlm.nih.gov), Swiss Prot (http://www.uniprot.org), the Protein family (Pfam) database (http://pfam.sanger.ac.uk), Cluster of Orthologous Groups databases (COG) (http://www.ncbi.nlm.nih.gov/COG), Gene Ontology (GO) (http://www.geneontology.org), and the Kyoto Encyclopedia of Genes and Genomes pathway database (KEGG) (http://www.genome.jp/kegg/pathway). GO and KEGG pathway enrichment analysis were carried out to further understand the biological functions of DEGs by using the Blast2GO and KOBAS2.0 programs, respectively [12, 13].

## qRT-PCR analysis

RNA were isolated from leaves with Trizol, and reverse transcription was performed with MMLV reverse transcriptase (TaKaRa, Dalian, China). qRT-PCR primers were designed with Primer Premier 6 (Table 1), and their specificity was verified by PCR. Expression of mRNA was determined by quantitative real-time PCR (qRT-PCR) using TIB8600 real-time PCR system (Triplex International Biosciences Co., LTD) and AceQ® qPCR SYBR® Green Master Mix (Vazyme) according to the manufacturers's instructions. The qRT-PCR reaction mixture

**Table 1. Specific primer pairs for genes from 'Yulu Xiang' Pear used in quantitative real-time polymerase chain reaction analysis.**

| Pathways | Genes name | Specific primers (5'to3') |
|---|---|---|
| Phenylpropanoid biosynthesis | Peroxidase | ATTTCAACGGCACTGGCAAC[a] |
| | | CTGATCGGTCTGGAGAAGCC[b] |
| | Caffeic acid 3-O-methyltransferase | AACTCACATCGGCACCAAGA[a] |
| | | GTGCAATCGAGGACGGAGTT[b] |
| | Shikimate O-hydroxycinnamoyltransferase | GCGTGGTTGCTAAACTGAGG[a] |
| | | CAAACCCACAACGGCTTTCC[b] |
| | Cinnamyl-alcohol dehydrogenase | CTTGGTGGTCTCGGACACTT[a] |
| | | TTAGCCAGGGCCGTCATTTC[b] |
| | Ferulate-5-hydroxylase | GACACTACGACTCCACCCAC[a] |
| | | CTTTTGGGAATGCGGTAGCC[b] |
| Cysteine and methionine metabolism | Methionine-gamma-lyase | GCCTGCAACATAACGACGAC[a] |
| | | GTATGTTGGCGACGGTGAGA[b] |
| | Cysteine synthase | GGATCTTCCTGCCACCCAAA[a] |
| | | CTAGCCGTGGGTTGGAACAT[b] |
| Glycometabolism | Malate dehydrogenase | AAACAACCAGCTCGCATCCT[a] |
| | | AGCATGATTCCCCTTGCGAT[b] |
| Monoterpenoid biosynthesis | (3S)-linalool synthase | AAGCGCGATGGTTTAGTTGC[a] |
| | | GCACGCCTGAACTGATGAAC[b] |
| Sesquiterpenoid and triterpenoid biosynthesis | Lupeol synthase 2 | GGTACGGGAATTGGGGTGTT[a] |
| | | CCCCAACCACCGTTCTCATT[b] |
| Ethylene synthesis | 1-aminocyclopropane-1-carboxylate synthase | AGTTATGAGTGGCGGAGGAG[a] |
| | | TGTCCCCTCACCGTAACATC[b] |
| | Aminocyclopropanecarboxylate oxidase | GCGACCTTCCCAGTTGTTGA[a] |
| | | AACCCCAGTTCTCACAAGCA[b] |
| Gibberellins biosynthesis | Gibberellin 2beta-dioxygenase | AAGTGTGAGGCACAGGGTTT[a] |
| | | GCTTTCCTCCTCTCCGTTCA[b] |
| Abscisic acid biosynthesis | 9-cis-epoxycarotenoid dioxygenase | CCAGACTCAATCCAAGGCGT[a] |
| | | GTAGCTGGCGGTTCCATCTT[b] |
| Reference gene | Tubulin | AGAACAAGAACTCGTCCTAC[a] |
| | | GAACTGCTCGCTCACTCTCC[b] |

[a] Forward primer.

[b] Reverse primer.

consisted of 10 μL of 2×SYBR real-time PCR premixture, 0.4 μl each forward and reverse primers (10 μM), 1 μL cDNA template and RNase free dH$_2$O to adjust to 20 μL. A reaction mixture of 20 μL was loaded into TIB8600 real-time PCR system and subjected to the following cycling: 5 min at 95˚C to denature DNA and activate Taq polymerase; 40 cycles of 15 s at 95˚C and 30 s at 60˚C. The Tubulin gene (AB239681) was amplified in parallel as an internal reference gene. Using the 2$^{-\Delta\Delta Ct}$ method [14], relative quantification of each gene mRNA expression was calculated by concurrent amplification of the Tubulin endogenous control. Three technical replicates were used for each sample and the data are shown as means ± standard deviation (S.D.).

## Statistical analysis

Data statistical analysis was carried out with SPSS 17.0 (SPSS, Chicago, IL, USA), and the one-way analysis of variance (ANOVA) and Student-Newman-Kreuls test at $p < 0.05$ were used for the statistical analysis between the control and drought stress groups.

## Results

### Effects of drought stress on multiple indicators of 'Yulu Xiang' Pear

Fruit quality analysis showed that the contents of fruit soluble solids in drought treatment groups (Fig 1A) were lower than that in control groups, and similar result about fruit weight

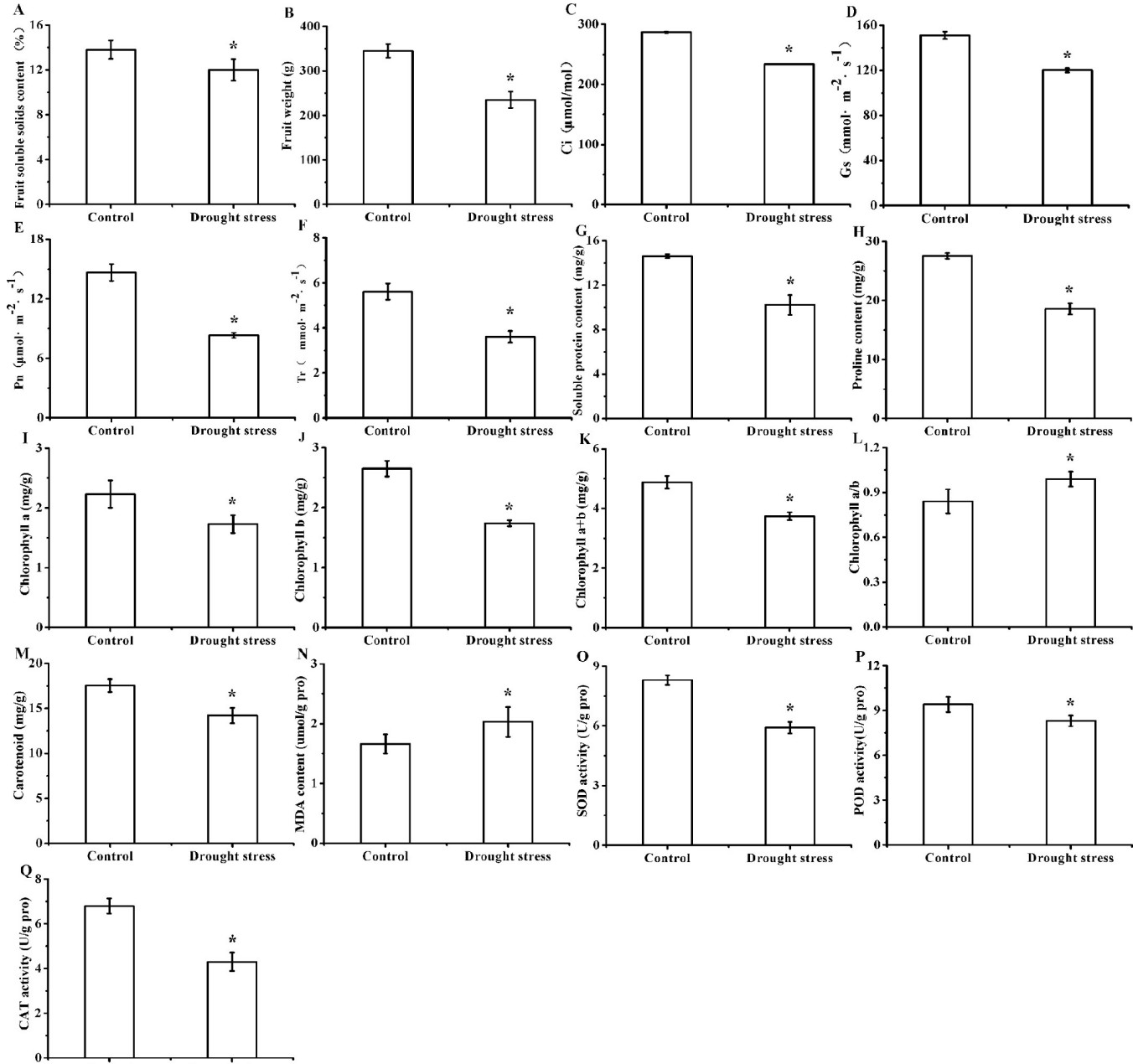

**Fig 1. Effects of long-term drought stress on fruit quality, photosynthetic indexes, antioxidant system, and osmotic adjustment system in leaves of 'Yulu Xiang' Pear.** Drought stress significantly decreased the content of fruit soluble solids (A), and reduced fruit weight (B) of 'Yulu Xiang' Pear. Photosynthetic characteristics including Ci (C), Gs (D), Pn (E), and Tr (F) in leaves of 'Yulu Xiang' Pear. Drought stress reduced the content of soluble protein (G) and proline (H) content. Levels of chlorophyll a (I), chlorophyll b (J), Chlorophyll a+b (K), Chlorophyll a/b (L), and carotenoids (M) in leaves of 'Yulu Xiang' Pear. Drought stress increased MDA (N) levels, but decreased the activities of SOD (O), POD (P), and CAT (Q). All data are expressed as mean ± standard deviation (n = 3), and statistical significance is denoted by $*$ between different groups ($P < 0.05$).

(Fig 1B) could be observed, indicating that the long-term drought stress negatively affected the nutrients accumulation and fruit development. The nutrients accumulation in fruit greatly depends on the transportation from photosynthesis products, and it was quite possible that the reduced fruit soluble solids may be resulted from the impaired photosynthesis in leaves induced by drought stress. The detection of photosynthetic characteristics showed that the Ci and Gs levels in leaves of drought treatment groups significantly decreased (Fig 1C and 1D). It indicated that drought stress weakened the capacities of carbon fixation and $CO_2$ exchange between plants and atmospheric environment, which partly resulted in the lower Pn and negatively affected the photosynthetic process in leaves (Fig 1E). The down-regulation of soluble protein (Fig 1G) and proline (Fig 1H) in leaves of drought treatment groups were also found, which further confirmed the inhibited photosynthetic process under drought stress. In plant stress tolerance system, soluble protein and proline are very important osmoregulation substance, and are closely related to the function of regulating osmotic pressure and protecting cell membrane structure. The reduction of soluble protein and proline suggested that plant regulation on osmotic pressure was incontrollable under drought stress, and the weakened tolerance negatively affected the capacity to absorb and/or retain water of 'Yulu Xiang' Pear which would result in the impairments of cell membrane structure. Fortunately, the stress protection in plants was still working. The down-regulated Gs blocked the process of water exchange, resulting in the reduced Tr levels in leaves under drought stress to reduce the the water loss rate (Fig 1F). Moreover, the photosynthetic process is closely related to the levels and composition of photosynthetic pigments. Compared to that in control groups, the levels of chlorophyll a, chlorophyll b, and carotenoid in leaves of drought stress groups were lower (Fig 1I, 1G, 1K and 1M), indicating that light utilization and photosynthetic productivity in leaves were seriously weakened by drought stress. However, the higher rates of chlorophyll a/b drought stress groups (Fig 1L) were observed, and the relative more content of chlorophyll a could enhance light utilization efficiency to improve photosynthesis under drought stress. The impairments of dought stress on photosynthetic pigments were mainly caused by the increased oxidative stress which was confirmed by the increased malondialdehyde (Fig 1N). The activities of SOD, POD, and CAT in leaves of drought stress treatment groups was down-regulated (Fig 1O–1Q), indicating that the weakened antioxidant systematization resulted in the enhanced oxidative stress under drought stress. These results showed that drought stress significantly reduced photosynthetic productivity, and seriously damaged physiological metabolism in leaves of 'Yulu Xiang' Pear.

## Read mapping

Transcriptomic raw data is available on Sequence Read Archive (SRA) database, and the SRA accession is PRJNA655255. According to the assembled pear genome of 'Dangshansuli' [15], reads of RNA-Seq data were mapped in this study. In total, two RNA-Seq libraries were sequenced from 'Yulu Xiang' Pear leaves of control and drought treatment groups, and there were 204,157,258 and 209,344,612 clean reads including 30,827,745,958 and 31,611,036,412 nucleotides in control and drought treatment groups, respectively (Table 2). The low-quality reads with connectors in above sequencing data would cause great interference to the subsequent information analysis, and it was necessary to further filter the sequencing data. Finally, 189,653,068 and 194,811,350 reads were obtained after splice and redundancy check.

## Differentially expressed genes (DEGs) and functional annotation of transcriptome sequencing based on public databases

In present study, differences of gene expression between control and drought treatment groups were analyzed. 2,207 differentially expressed genes (DEGs) with expression fold change

**Table 2. Summary of read numbers based on the RNA-Seq data from 'Yulu Xiang' Pear leaves.**

| Summary | Control | Drought stress |
|---|---|---|
| Reads No. | 204,157,258 | 209,344,612 |
| Bases (bp) | 30,827,745,958 | 31,611,036,412 |
| Q30 (bp) | 28,252,595,080 | 24,352,878,562 |
| N (%) | 0.0043574 | 0.0044588 |
| Q20 (%) | 96.316 | 96.542 |
| Q30 (%) | 91.644 | 92.066 |
| Clean Reads No. | 189,653,068 | 194,811,350 |
| Clean Data (bp) | 28,637,613,268 | 29,416,513,850 |
| Clean Reads % | 92.888 | 93.050 |
| Clean Data % | 92.888 | 93.050 |

Note: Reads No.: total number of reads; Bases (bp): total number of bases; Q30 (bp): the total number of bases whose recognition accuracy is more than 99.9%; N (%): percentage of fuzzy bases; Q20 (%): the percentage of bases whose recognition accuracy is more than 99%; Q30 (%): percentage of bases with accuracy over 99.9%; Clean reads No: number of high-quality sequence reads; Clean data (bp): base number of high quality sequence; Clean reads %: the percentage of high-quality sequence reads in sequencing reads; Clean data %: the percentage of high-quality sequence bases to sequence bases.

of at least 2 fold up or down-regulation (FDR < 0.05) were identified from RNA-Seq by DESeq. To clear the functions of these DEGs, Gene Ontology (GO) term enrichment analysis were performed to classify the functions of DEGs in different categories. Based on sequence homology, the assigned GO terms were summarized into the three main GO categories, biological process (BP), cellular component (CC), and molecular function (MF), and then into 88 main functional categories including biological process, oxidation-reduction process, microtubule-based process, and etc (Fig 2). According to GO annotation and functional enrichment analysis of DEGs, biological process comprised 45 GO annotations (51.14%) and was the largest one, followed by molecular function 25 GO annotations (28.41%), and cellular component 18 GO annotations (20.45%). Among, 290 and 55 DEGs involved in biological process and oxidation-reduction process, indicating that the synthesis of metabolites and antioxidant capacity in leaves were significantly affected by drought stress.

The metabolic pathway $P \leq 0.05$ was defined as the Kyoto Encyclopedia of Genes and Genomes (KEGG) pathway with significant enrichment of DEGs using a BLAST search against the KEGG database. 58 DEGs involved in biosynthesis of secondary metabolites were found (Table 3), which was consistent with results through GO term enrichment analysis. 17, 12, 8, 6, 6, 5, and 4 DEGs were annotated from phenylpropanoid biosynthesis pathway, cysteine and methionine metabolism pathway, flavonoid biosynthesis pathway, diterpenoid biosynthesis pathway, carotenoid biosynthesis pathway, monoterpenoid biosynthesis pathway, sesquiterpenoid and triterpenoid biosynthesis pathway and brassinosteroid biosynthesis pathway, respectively. To validate the reliability of the drought responsive gene expression profiles for DEGs, 16 genes encoding Peroxidase, Caffeic acid 3-O-methyltransferase, Shikimate O-hydroxycinnamoyltransferase, Cinnamyl-alcohol dehydrogenase, Cinnamoyl-CoA reductase, Ferulate-5-hydroxylase, aminocyclopropane-1-carboxylate synthase, Aminocyclopropanecarboxylate oxidase, Methionine-gamma-lyase, Cysteine synthase, Malate dehydrogenase, Gibberellin 2beta-dioxygenase, cis-epoxycarotenoid dioxygenase, and (3S)-linalool synthase, Lupeol synthase 2 were randomly selected and confirmed by quantitative real-time PCR using gene-specific primers. Even there was the difference in the fold change between the two

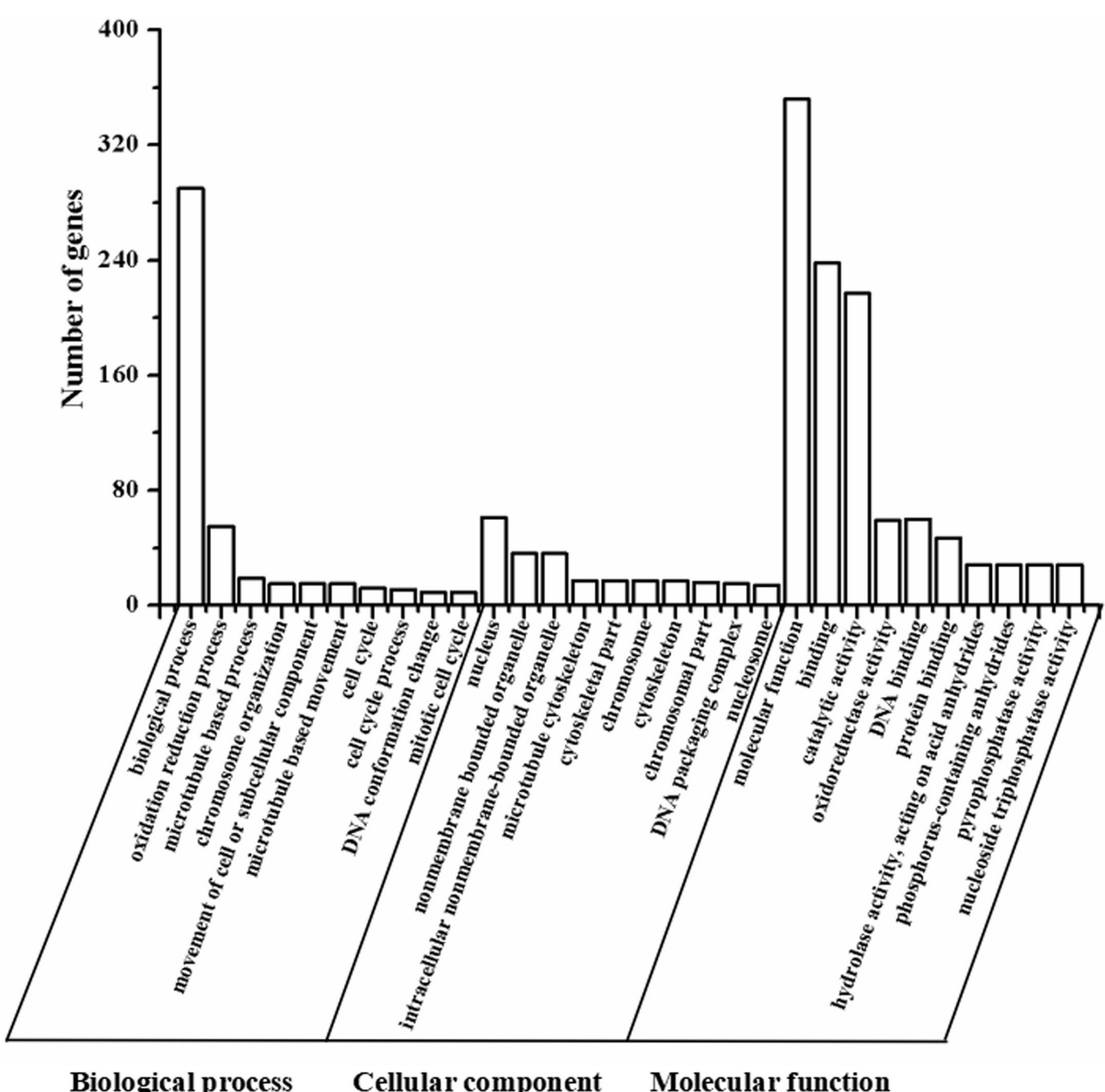

**Fig 2. Summary of deferentially expressed genes in leaves of 'Yulu Xiang' Pear under drought stress by gene ontology classification.** The genes were annotated into three main categories: biological process, cellular component and molecular function (X-axis). Y-axis (left) represents the number of deferentially expressed genes.

methods, results of relative expression patterns from the drought stress responsive genes carried out by quantitative real-time PCR were consistent with these from DEGs data (Fig 3). The high correlation in relative expression patterns between qPCR and RNA-Seq suggests that the present DEGs data are reliable, which could be an efficient tool for further study to investigate the functions of drought stress responsive genes.

**Table 3. Summary of deferentially expressed genes in leaves of 'Yulu Xiang' Pear under drought stress by KEGG classification.**

| Pathway ID | Pathway | DEG number | UP regulated | DOWN regulated | FDR |
|---|---|---|---|---|---|
| ko00940 | Phenylpropanoid biosynthesis | 17 | 15 | 2 | 0.008 |
| ko00270 | Cysteine and methionine metabolism | 12 | 5 | 7 | 0.002 |
| ko00941 | Flavonoid biosynthesis | 8 | 8 | 0 | 0.001 |
| ko00904 | Diterpenoid biosynthesis | 6 | 4 | 2 | 0.002 |
| ko00906 | Carotenoid biosynthesis | 6 | 6 | 0 | 0.006 |
| ko00902 | Monoterpenoid biosynthesis | 5 | 5 | 0 | 0.000 |
| ko00909 | Sesquiterpenoid and triterpenoid biosynthesis | 4 | 3 | 1 | 0.020 |

## Discussion

Drought stress, known as water deficit stress, is the most common environmental stress with many characteristics including high frequency, wide distribution, long duration and great harm. Drought stress negatively affects normal physiological function and development of plants, and seriously threatens agricultural production. Present study found that long-term drought stress decreased the levels of fruit soluble solids and fruit weight, and caused seriously impairments on the fruit quality of 'Yulu Xiang' Pear (Fig 1A and 1B). The similar results could be found in other plants, such as strawberry, cantaloupe, and cotton [16–18], suggesting that the nutrients accumulation and fruit development were severely inhibited by drought stress. The nutrients accumulation in fruit are closely related with photosynthesis in leaves [19]. In present study, the down-regulation of Ci and Gs was observed. The lower intercellular $CO_2$ concentration and blocked $CO_2$ exchange between plants and atmospheric environment would reduce carbon fixation, and result in a lower Pn levels (Fig 1E), and finally block photosynthetic process in leaves [20]. Moreover, we also found that the levels of soluble protein and proline in leaves of drought treatment groups was reduced, which would negatively affect the stress tolerance system in leaves. In plant stress tolerance system, soluble protein and proline are very important osmoregulation substance, and play the key role in regulating osmotic pressure and protecting cell membrane structure [21]. The reduction of soluble protein and proline under long-term drought stress could cause that the plant regulation on osmotic pressure was seriously impaired, which would negatively affect the capacity to absorb water in leaves of 'Yulu Xiang' Pear and finally induce oxidative stress.resulting in the impairment of cell membrane structure. Fortunately, plants have gradually formed unique growth habits and physiological characteristics in respond to drought stress during the long-term evolution process [22]. Under drought stress, the down-regulated Gs blocked the process of water exchange between plants and atmospheric environment. The reduced Tr and slowed photosynthesis process in 'Yulu Xiang' Pear leaves could effectively reduce water loss to adapt to drought conditions.

The photosynthetic process also depends on the the contents and composition of photosynthetic pigments [23]. Our study found that the levels of chlorophyll a, chlorophyll b, and carotenoid in drought stress groups were lower than that in control groups, and the weakened light utilization indicated a reduced photosynthetic productivity in leaves. However, a higher rates of chlorophyll a/b were also observed in leaves of drought stress groups, and it indicated a relatively more chlorophyll a which was helpful to increase light utilization efficiency and improve photosynthesis to adapt to the drought stress [24]. The impairments of long-term dought stress on photosynthetic pigments were closely related to the oxidative stress [25], which was confirmed by the increased level of lipid peroxidation in drought treatment groups (Fig 1N). Sharma et al. [26] found that the oxidative stress induced by mild drought stress could

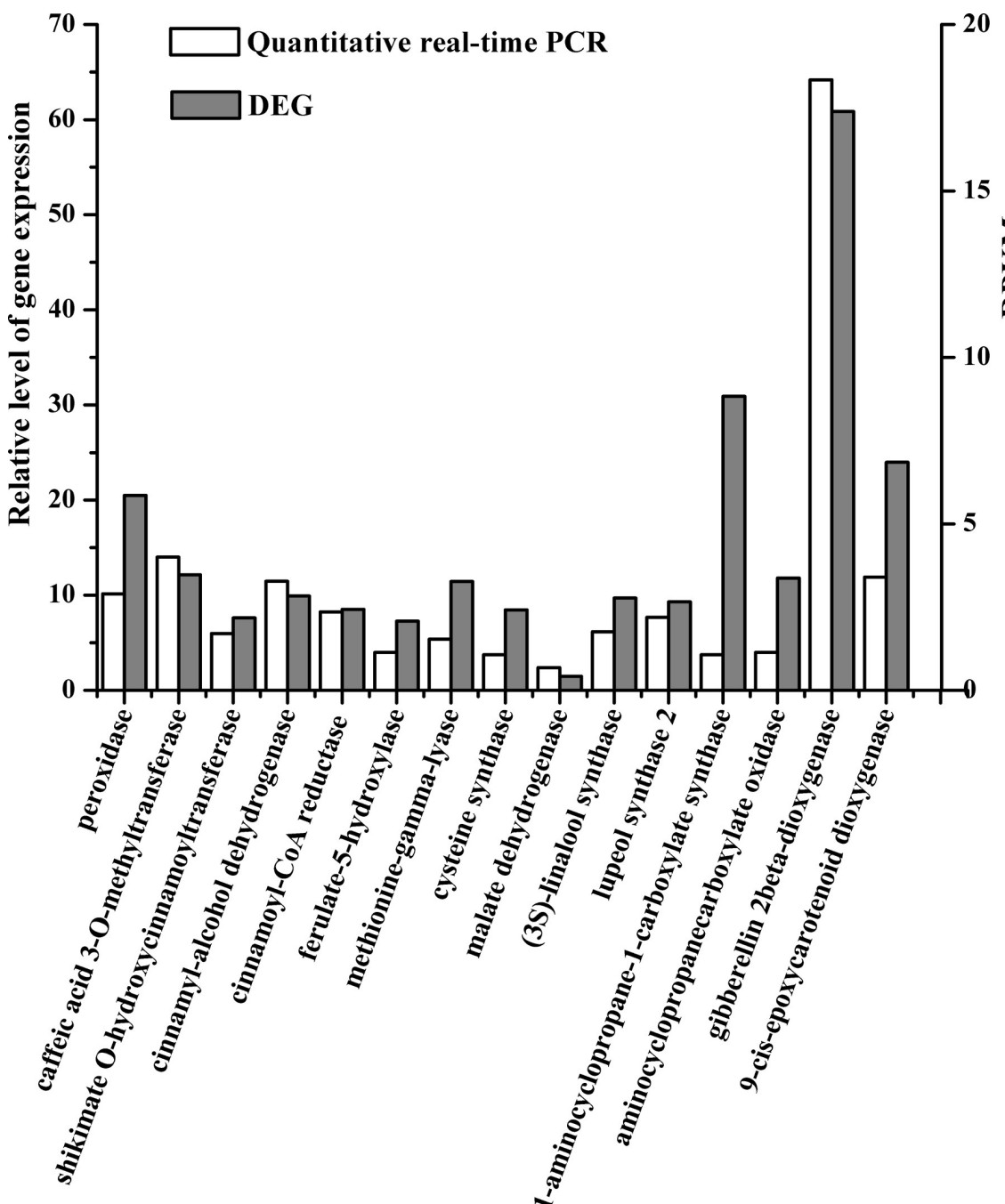

**Fig 3. Reliability and consistency verification of DGE by quantitative real time PCR.** The x-axis represents individual genes and the y-axis fold-change in expression by RPKM method (gray bars) or qRT-PCR (white bars).

promote the antioxidant activities, but the concentration glutathione and the activities of total superoxide dismutases (SODs) were inhibited under severe drought stress. The results of our study were similar to those found, and the activities of SOD, POD, and CAT in leaves under long-term drought stress was down-regulated (Fig 1O–1Q), indicating that long-term drought stress severely impaired the antioxidant systematization and the damage induced by increased oxidative stress was uncontrollable.

Studies on the drought resistance mechanism of plants found variety of molecular mechanisms including up-regulating the cell osmotic regulation ability by the induced synthesis of aquaporin and small molecule soluble organic matter, increasing the stability of biomembrane and protein by induced synthesis of LEA protein and membrane protein, clearing free radicals by induced antioxidant protection enzyme system under moderate drought stress [27, 28]. However, we found a reduced soluble protein and proline in leaves of 'Yulu Xiang' Pear, and the down-regulated osmotic potential indicated a greatly damaged ability of cell to absorb or retain water. The severe oxidative stress induced by the lack of water in plants could severely impair cell membrane system, and negatively affect the photosynthetic pigment synthesis, finally resulting in the reduction of light utilization and photosynthetic productivity in leaves. A vicious circle involving osmotic pressure and photosynthate was created under drought stress (Fig 4).

Drought stress could induce the accumulation of secondary metabolites (such as terpenes and flavonoids) in plant roots to enhance osmotic regulation and clear free radicals to adapt the drought environment [29, 30]. Indeed, the analysis of KEGG pathway enrichment of DEGs in present study showed that drought stress up-regulated the expression of key enzyme genes related to terpenoids, flavonoids, methionine, and cysteine biosynthesis pathway (Table 2), indicating that drought stress induced these substances synthesis to enhance osmotic stress and relieve oxidative stress in 'Yulu Xiang' Pear leaves. The osmotic regulation and antioxidant capacity of leaves in drought treatment groups were achieved by up-regulated expression of malate dehydrogenase, indicating a tolerance to a drought stress by the promoted biosynthesis of malate and soluble sugar. Similar results were found in wheat investigated by Cui et al. [31]. The soluble sugar not only could regulate osmotic stress and relive free radicals, but also provide energy for the basic physiological function of leaves cells by respiration [32]. We also found that drought stress increased the gene expression of limiting enzymes phenylpropanoid biosynthesis including peroxidase, caffeic acid 3-O-methyltransferase, shikimate O-hydroxycinnamoyltransferase, cinnamyl-alcohol dehydrogenase, cinnamoyl-CoA reductase, ferulate-5-hydroxylase. The enhanced phenylpropanoid biosynthesis would be helpful to the synthesis of lignin, indicating that cell wall synthesis was promoted under drought stress to strengthen the water transportation system and osmotic pressure in leaves [33].

Plant hormone is an important factor regulating various physiological processes including cell division, elongation, differentiation, and plant germination, rooting, flowering, fruiting, sex determination, dormancy and abscission [34]. The DEGs analysisin transcriptome of 'Yulu Xiang' Pear leaves showed significant effects of drought stress on ethylene, abscisic acid, gibberellin, brassinolide and other plant hormone biosynthesis or signal transduction pathways. Among, the mRNA expression levels of key enzyme genes related to ethylene biosynthesis (1-aminocyclopropane-1-carboxylic acid synthetase, and aminocyclopropanecarboxylate oxidase) and abscisic acid biosynthesis (9-cis-epoxycarotenoid dioxygenase) were promoted, and these hormones could significantly quicken cell division and leaf abscission to reduce transpiration rate and improve the water use efficiency of plants during drought stress. Arraes et al. [35] also found that the expression multiple genes involved in the biosynthesis of phytohormones including ethylene and ABA were up-regulated under drought stress, and responses to drought stress could be regulated by a crosstalk network among phytohormones signaling pathways.

## Conclusion

Present study found that long-term drought stress caused the reduction of nutrients accumulation and inhibition of fruit development. The impairments of the fruit quality of 'Yulu Xiang'

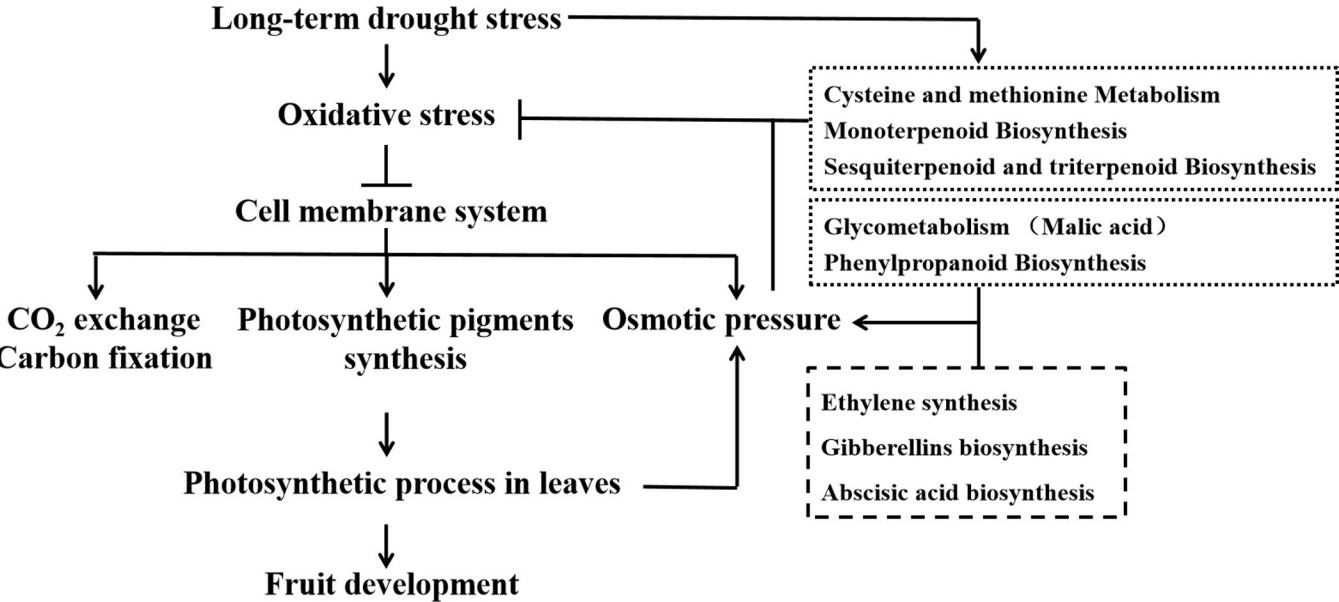

**Fig 4. The model proposing the effects and adaptive mechanism of 'Yulu Xiang' Pear in response to long-term drought stress.** Arrows (→) and truncated lines (—|) indicated promoting and inhibiting effects, respectively.

Pear mainly resulted from the inhibited antioxidant systematization and increased oxidative stress induced by drought stress. Reduced protective effects against oxidative stress inhibited the synthesis of photosynthetic pigment, and reduced light utilization and photosynthetic productivity, finally negatively affected the osmotic pressure. All of the data presented here suggests that a network consisting of multiple secondary metabolites and phytohormones signaling pathways is involved in the regulation of adaptive mechanism of 'Yulu Xiang' Pear in response to long-term drought stress (Fig 4).

## Author Contributions

**Conceptualization:** Guowei Hao, Huangping Guo.

**Formal analysis:** Sheng Yang, Xiaowei Zhang.

**Funding acquisition:** Huangping Guo, Baochun Fu.

**Investigation:** Sheng Yang, Xiaowei Zhang.

**Methodology:** Sheng Yang, Mudan Bai, Guowei Hao, Xiaowei Zhang.

**Project administration:** Sheng Yang.

**Writing – original draft:** Sheng Yang, Huangping Guo.

**Writing – review & editing:** Huangping Guo, Baochun Fu.

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
