## [Decision Letter · Decision Letter 0]

10 Dec 2020

PONE-D-20-32528

Transcriptome survey and expression  analysis  reveals effects and metabolic pathways of 'Yulu Xiang' Pear in response to long-term drought stress

PLOS ONE

Dear Dr. Guo,

Thank you for submitting your manuscript to PLOS ONE. After careful consideration, we feel that it has merit but does not fully meet PLOS ONE’s publication criteria as it currently stands. Therefore, we invite you to submit a revised version of the manuscript that addresses the points raised during the review process.

The reviewers raised some comments and concerns, please address the comments accordingly. 

We look forward to receiving your revised manuscript.

Kind regards,

Xiang Jia Min, Ph. D.

Academic Editor

PLOS ONE

Journal Requirements:

2. Please include your tables as part of your main manuscript and remove the individual files. Please note that supplementary tables should be uploaded as separate "supporting information" files.

Reviewers' comments:

Reviewer's Responses to Questions

**Comments to the Author**

1. Is the manuscript technically sound, and do the data support the conclusions?

Reviewer #1: Partly

Reviewer #2: Yes

2. Has the statistical analysis been performed appropriately and rigorously? 

Reviewer #1: Yes

Reviewer #2: Yes

3. Have the authors made all data underlying the findings in their manuscript fully available?

Reviewer #1: Yes

Reviewer #2: Yes

4. Is the manuscript presented in an intelligible fashion and written in standard English?

Reviewer #1: Yes

Reviewer #2: Yes

5. Review Comments to the Author

Reviewer #1: The article “Transcriptome survey and expression analysis reveals effects and metabolic pathways of 'Yulu Xiang' Pear in response to long-term drought stress” measures some physiological indicator and has a RNA-Seq on the drought stress pears. I have some suggestions as follows.

1. Combine figure1-figure5 into one figure.

2. Please present the useful information of RNA-Seq in the main text.

3. In Table 2, deferentially is differentially.

4. How many repetitions of the RNA-Seq and other data.

5. Please describe a model in the end of the text based on your data.

6. The results are simple, rewrite it.

Reviewer #2: Though Transcriptome sequencing-based works are very common nowadays for investigating the role, metabolic pathways, the discovery of candidate genes and markers; the manuscript entitled “Transcriptome survey and expression analysis reveals effects and metabolic pathways of 'Yulu Xiang' Pear in response to long-term drought stress” is important by different aspects, one the Pear crop is economically important as its fruits have high demands worldwide and another is the drought stress which causes the serious problems in its production. In the current work, many metabolic pathways are depicted from the transcriptome sequence of the Yulu Xiang Pear, which have major importance in the abiotic stress tolerance in the plant. I appreciate the efforts of the authors and would like to add comments on this research work for some clarity on the following points.

• The authors had mention the water stress condition created for the 12 months by withholding irrigation in the drought stress group, however, there is no evidence shown in the result to monitored the stress level for this large period. They could have to monitor the relative water content, chlorophyll content, cell membrane stability periodically.

• The authors have mentioned the soil relative water content in Pear trees cultivation was achieved 80% and 30% in control and drought stress groups respectively, but it is not cleared that at what stage the said moisture levels were achieved and for how much period these levels were maintained to induce the stress in pear trees.

• The sources for the software packages and databases used for sequence analysis and annotation were not mentioned in the manuscript. The sources/links should be cited or acknowledged in the manuscript.

• The functions of the selected genes for the quantitative real-time PCR analysis were not discussed in the manuscript, on what basis the gene was selected for validation? The candidate genes for drought stress in Pear plant should be discussed in the discussion part of the manuscript correlating with the antioxidant enzymes assay, fruit soluble solids, and photosynthetic characters performed in the current research work.

• Proline content is expected to be increase during drought stress conditions but in the current study, it is lower in drought stress than the control group. Please clarify it in the discussion with reference as it is a contrasting result.

• The antioxidant enzymes SOD, CAT, POD activities reported decreased under drought stress as compared to the control group. At initial stress, these antioxidant enzyme levels should increase but might be because of the long-term moisture stress condition these enzyme activities are reduced. Please give references for the same. Also, the authors why not monitored the enzyme activity during stress conditions periodically so that the stage at which the activity is at its pick level could have detected.

• The changes in fruit soluble solid content and quality of fruit were checked in the current work, have there any impact of drought stress on fruit-bearing and yield were not mentioned, the yield and production-related data of the control and drought stress groups should be included in the results as it directly indicates the stress impact on the production/yield from that plant.

• The discussion part of the manuscript has to improve by including some additional references.

Besides these points, there are some minor typo and grammatical errors are there which need to be corrected.

L 172 – Add “and” in-between two values as it creating the confusions.

L 174 - L 176- Modify the sentence for more clarity.

L 210 - Is the author wants to say “for further study to invest ‘investigate’ the functions of drought stress-responsive genes”? Correct the sentence.

L 231 – L 232- Reference is missing for the statement.

L 258 - L 260- Add reference for the sentence.

Fig. 6 & Fig. 7- The labels of the X-axis are difficult to read, use good quality graphical image.

6. PLOS authors have the option to publish the peer review history of their article (what does this mean?). If published, this will include your full peer review and any attached files.

Reviewer #1: No

Reviewer #2: **Yes: **Dr. Pranjali Atul Gedam

---

## [Author Response · Author response to Decision Letter 0]

29 Dec 2020

Responses to Reviewers

Dear Editor

We want to thank you and 2 reviewers for your useful comments and suggestions on our manuscript. We have modified the manuscript, added more information accordingly, and detailed corrections are listed below point by point:

 AUTHOR RESPONSE: 

 Thank you for your suggestions, and modifications including format among others of Abstract, Introduction, Materials and Methods have been made according to PLOS ONE's style requirements. 

2）Please include your tables as part of your main manuscript and remove the individual files. Please note that supplementary tables should be uploaded as separate "supporting information" files.

AUTHOR RESPONSE:

Thanks for your comments. The tables and have been included as part of manuscript. The supplementary files have been deleted, and is moved to the paper as as part of manuscript. Thanks again. 

REVIEWER 1: 

Reviewer #1: The article “Transcriptome survey and expression analysis reveals effects and metabolic pathways of 'Yulu Xiang' Pear in response to long-term drought stress” measures some physiological indicator and has a RNA-Seq on the drought stress pears. I have some suggestions as follows.

1.Combine figure1-figure5 into one figure.

AUTHOR RESPONSE:

Thanks for your useful suggests. The Figure 1-5 have been combined into one figure. The corresponding information in results are modified, and the new figure can be found in attached files . Thanks again.

2.Please present the useful information of RNA-Seq in the main text.

AUTHOR RESPONSE:

Thanks for your comments. The information about RNA-Seq have been rewritten , and modified information could be found in the results of new manuscript, which have been noted in the Results with red text. Thanks again for your suggestions. 

3.How many repetitions of the RNA-Seq and other data.

AUTHOR RESPONSE: 

We are sorry for our mistakes. There were six replicates for each treatment used for the RNA-Seq, and the experiments about fruit quality, photosynthetic indexes, antioxidant system were repeated three times. The above content have been added in the new paper. Thanks again for your comments.

4.Please describe a model in the end of the text based on your data.

AUTHOR RESPONSE: 

Thanks for your suggestions. Basing on the results of biochemical indexes and transcriptome sequencing, we made a description containing words and model figure about the adaptive mechanism of 'Yulu Xiang' Pear in response to long-term drought stress. The added contents could be found in paper in the conclusion of paper. Thanks again.

5.The results are simple, rewrite it.

AUTHOR RESPONSE: 

Thanks for your comments. Indeed, the results are too simple to describe our works. Therefor, Results are rewritten and modified, and more contents are added into the Results, Discussion, and Conclusion. Thanks again for your useful suggestions.

REVIEWER 2: 

1.The authors had mention the water stress condition created for the 12 months by withholding irrigation in the drought stress group, however, there is no evidence shown in the result to monitored the stress level for this large period. They could have to monitor the relative water content, chlorophyll content, cell membrane stability periodically.

AUTHOR RESPONSE: 

Thanks a lot for your suggestions. Undoubtedly, your comments are absolutely correct. The detection of water content in soil or other indexes should be carried out every three or four months, which is more reasonable and reliable. However, we just take one test after 12 months, and it is an irreparable mistakes in experimental design. We will take your suggestion in future study, and make our effort to avoid similar mistakes. Thanks again for your useful suggestions.

2.The authors have mentioned the soil relative water content in Pear trees cultivation was achieved 80% and 30% in control and drought stress groups respectively, but it is not cleared that at what stage the said moisture levels were achieved and for how much period these levels were maintained to induce the stress in pear trees.

AUTHOR RESPONSE: 

We are so sorry for the missing description in Test Design. In fact, the pear tress in control group were given reasonable irrigation as follows: Pear trees were watered for the first time after fertilization in autumn (October 10, 2017) , and the ground was also covered with plastic film to save water in winter. Another three watering was carried out on March 5th( 2018), May 15th (2018), and July 1st (2018), respectively. For drought treatment groups, there was no water spray in the orchard. 

Based on the above description, we think that the long term drought stress lasted for one year. However, the detection of soil relative water content obtained on October 21 (2018) could not fully prove the drought stress on tress during the whole year. As the response to your first question, more detection about the soil relative water content in different time is more reasonable. This is still an irreparable mistakes during the experiments. Thanks again for your comments.

3.The sources for the software packages and databases used for sequence analysis and annotation were not mentioned in the manuscript. The sources/links should be cited or acknowledged in the manuscript.

AUTHOR RESPONSE: 

Thanks for your suggestion. We really appreciate all the help from the software and databases, and more related information are added in the Method. Thanks again for your suggestion.

4.The functions of the selected genes for the quantitative real-time PCR analysis were not discussed in the manuscript, on what basis the gene was selected for validation? The candidate genes for drought stress in Pear plant should be discussed in the discussion part of the manuscript correlating with the antioxidant enzymes assay, fruit soluble solids, and photosynthetic characters performed in the current research work.

Proline content is expected to be increase during drought stress conditions but in the current study, it is lower in drought stress than the control group. Please clarify it in the discussion with reference as it is a contrasting result.

The antioxidant enzymes SOD, CAT, POD activities reported decreased under drought stress as compared to the control group. At initial stress, these antioxidant enzyme levels should increase but might be because of the long-term moisture stress condition these enzyme activities are reduced. Please give references for the same. Also, the authors why not monitored the enzyme activity during stress conditions periodically so that the stage at which the activity is at its pick level could have detected.

The changes in fruit soluble solid content and quality of fruit were checked in the current work, have there any impact of drought stress on fruit-bearing and yield were not mentioned, the yield and production-related data of the control and drought stress groups should be included in the results as it directly indicates the stress impact on the production/yield from that plant.

The discussion part of the manuscript has to improve by including some additional references.

AUTHOR RESPONSE: 

According to your suggestion, more information about the functions of the selected genes and relation between candidate genes with antioxidant enzymes assay, fruit soluble solids, and photosynthetic characters are added in the part of Discussion and conclusion. And, more related literatures are cited. Thanks again for your suggestions, and detailed information can be found in newly modified articles.

5.Besides these points, there are some minor typo and grammatical errors are there which need to be corrected.

L 172 – Add “and” in-between two values as it creating the confusions.

L 174 - L 176- Modify the sentence for more clarity.

L 210 - Is the author wants to say “for further study to invest ‘investigate’ the functions of drought stress-responsive genes”? Correct the sentence.

L 231 – L 232- Reference is missing for the statement.

L 258 - L 260- Add reference for the sentence.

Fig. 6 & Fig. 7- The labels of the X-axis are difficult to read, use good quality graphical image.

AUTHOR RESPONSE: 

Thank your for your suggestion and comments. The incorrect or incomprehensible sentences have been modified, and relative literature is cited. The figure is replaced with another good quality graphical image, and more information could be found in newly modified articles. Thanks again for your suggestions

---

## [Decision Letter · Decision Letter 1]

13 Jan 2021

Transcriptome survey and expression analysis reveals the adaptive mechanism of 'Yulu Xiang' Pear in response to long-term drought stress

PONE-D-20-32528R1

Dear Dr. Guo,

We’re pleased to inform you that your manuscript has been judged scientifically suitable for publication and will be formally accepted for publication once it meets all outstanding technical requirements.

Kind regards,

Xiang Jia Min, Ph. D.

Academic Editor

PLOS ONE

Additional Editor Comments (optional):

Reviewers' comments:

Reviewer's Responses to Questions

**Comments to the Author**

1. If the authors have adequately addressed your comments raised in a previous round of review and you feel that this manuscript is now acceptable for publication, you may indicate that here to bypass the “Comments to the Author” section, enter your conflict of interest statement in the “Confidential to Editor” section, and submit your "Accept" recommendation.

Reviewer #1: All comments have been addressed

Reviewer #2: All comments have been addressed

2. Is the manuscript technically sound, and do the data support the conclusions?

Reviewer #1: Partly

Reviewer #2: Yes

3. Has the statistical analysis been performed appropriately and rigorously? 

Reviewer #1: Yes

Reviewer #2: Yes

4. Have the authors made all data underlying the findings in their manuscript fully available?

Reviewer #1: Yes

Reviewer #2: Yes

5. Is the manuscript presented in an intelligible fashion and written in standard English?

Reviewer #1: Yes

Reviewer #2: Yes

6. Review Comments to the Author

Reviewer #1: Drought has seriously affected crop production, but the regulatory network of crop stress resistance is not clear,

There is no systematic theory to guide breeding. This paper give some useful informaton. My questions was all addressed, i have no more suggestions.

Reviewer #2: All comments have been address by authors. I want to suggest the authors, the experiment should be designed prior to start the experimental activities, So the observations will not missed on specific stage, which is very important in fruit trees. As the spel of experiment is very long, and it is hard to repeat the experimental activities.

7. PLOS authors have the option to publish the peer review history of their article (what does this mean?). If published, this will include your full peer review and any attached files.

Reviewer #1: No

Reviewer #2: **Yes: **Dr. Pranjali Atul Gedam

---

## [Editor Report · Acceptance letter]

25 Jan 2021

PONE-D-20-32528R1 

Transcriptome survey and expression analysis reveals the adaptive mechanism of 'Yulu Xiang' Pear in response to long-term drought stress 

Dear Dr. Guo:

I'm pleased to inform you that your manuscript has been deemed suitable for publication in PLOS ONE. Congratulations! Your manuscript is now with our production department. 

Kind regards, 

on behalf of

Dr. Xiang Jia Min 

Academic Editor

PLOS ONE